# Engineered Superabsorbent Nanocomposite Reinforced with Cellulose Nanocrystals for Remediation of Basic Dyes: Isotherm, Kinetic, and Thermodynamic Studies

**DOI:** 10.3390/polym14030567

**Published:** 2022-01-30

**Authors:** Arej S. Al-Gorair, Asmaa Sayed, Ghada A. Mahmoud

**Affiliations:** 1Chemistry Department, College of Science Princess, Nourah bint Abdulrahman University, Riyadh 11564, Saudi Arabia; asalgorir@pnu.edu.sa; 2Polymer Chemistry Department, National Center for Radiation Research and Technology, Egyptian Atomic Energy Authority, Nasr City, P.O. Box 29, Cairo 11787, Egypt; ghadancrrt@yahoo.com

**Keywords:** cellulose nanocrystals, gamma radiation, isotherm study, kinetics study, nanocomposite

## Abstract

In this study, cellulose nanocrystals (CNCs) were produced from pea peels by acid hydrolysis to be used with pectin and acrylic acid (AAc) to form Pectin-PAAc/CNC nanocomposite by γ-irradiation. The structure, morphology, and properties of the nanocomposite were investigated using Fourier transform infrared spectroscopy (FTIR) and atomic force microscopy (AFM) techniques. The nanocomposite hydrogel was used for the removal of methylene blue dye (MB) from wastewater. The results revealed that the presence of CNCs in the polymeric matrix enhances the swelling and adsorption properties of Pectin-PAAc/CNC. The optimum adsorbate concentration is 70 mg/L. The kinetic experimental data were fit by pseudo-first-order (PFO), pseudo-second-order (PSO), and Avrami (Avr) kinetic models. It was found that the kinetic models fit the adsorption of MB well where the correlation coefficients of all kinetic models are higher than 0.97. The Avr kinetic model has the lowest ∆q_e_ (normalized standard deviation) value, making it the most suitable one for describing the adsorption kinetics. The adsorption isotherm of MB by Pectin-PAAc follows the Brouers–Sotolongo model while that by Pectin-PAAc/CNC follows the Langmuir isotherm model. The negative values of ∆G confirmed the spontaneous nature of adsorption, and the positive value of ∆H indicated the endothermic nature of the adsorption.

## 1. Introduction

Green chemistry is a necessity for improving human society. It is used to reduce toxic components from the environment. Water contamination is a global issue that creates serious health problems [1]. Dye pollution has been given much consideration; it damages natural ecosystems by discharging unlimited dyes into wastewater. The dye molecules are stable and challenging to biodegrade under normal conditions, and they are toxic, carcinogenic, and allergenic to the human body [2]. Around 3 × 10^5^ tons of synthetic dyes are employed in the textile industry every year [3]. Cationic (basic) and anionic (acidic) dyes are the most commonly used dyes in the textile industry because of their excellent water solubility and relatively low cost [4,5]. Basic dyes contain monoazoic, diazoic, and azine compounds [6]. Among various basic dyes, methylene blue (MB) is the most common. It is used for many purposes, such as printing, paper coloring, wool dyeing, cotton dyeing, tannin dyeing, as a temporary hair colorant, and as an indicator in the oxidation-reduction process [7]. Many technologies have been employed to extract dyes from wastewater, such as flocculation, biological treatment, electrochemical treatment, membrane filtration, ion exchange, and adsorption [8,9]. Adsorption provides excellent results because it is economical, effortless to produce, inconsiderate to toxic substances, and efficient for different dyes categories [10].

Polymer-based materials are widely used to adsorb a wide range of water contaminants due to their easy processability and tailor-made properties [11]. Superabsorbents are a loosely crosslinked network of hydrophilic polymers that can absorb and retain a lot of aqueous fluids, and the absorbed water is hardly removable even under some pressure [12]. A great approach is producing superabsorbent materials based on natural polymers. They have high biocompatibility, biodegradability, and water uptake capacity, as well as low toxicity [13]. The hydrophilic groups responsible for the high swelling of the polymeric chains are carboxyl, amino, amide, hydroxyl, and sulfonic [14].

Pectin is a non-toxic, biodegradable, and biocompatible polymeric material [15]. It is found in the peel and pulp of several fruits and is mainly prepared from citrus peel and apple pomace [16]. Pectin is an anionically charged structural plant polysaccharide consisting of a linear backbone of (1-4)-D-galacturonic acid residue with a few 1,2-D-rhamnose residues in the main chain, and some D-galactose and D-arabinose as side chains [17]. The degree of esterification depends on the source from which the pectin is isolated. It is widely used as a gelling, thickening, and stabilizing agent in the food industry [18]. Pectin has limited thermal stability and mechanical properties. However, it is blended with different polymers in composites to overcome these issues [19].

Cellulose nanocrystal (CNC) is a highly crystalline nanoparticle obtained by the acid hydrolysis of native cellulose by the removal of the amorphous phase [20]. CNCs are attractive for use as a filler in the hydrogel matrix due to their renewability, abundance, cytocompatibility, and excellent mechanical properties [21,22]. The presence of hydroxyl groups on the surface is responsible for reactivity, hydrophilicity, and the cellulose’s physical properties [23,24]. Those traits support the use of cellulose-based hydrogel in water treatment [25]. Many studies used CNC composite-based hydrogels in dye absorption [26,27]. Chaichi et al. [28] used crystalline nanocellulose (CNC) for pectin film. The results showed a positive relationship between the CNC level (2%, 5%, and 7% *w*/*w*) and degree of crystallinity. It was concluded that 5% CNC/pectin film can be considered an effective strengthening strategy in the use of pectin-based materials for food packaging applications.

In this study, Pectin-PAAc/CNC nanocomposite hydrogel was prepared using the gamma irradiation technique. This technique has the advantage of liberty from toxic impurities such as crosslinking agents and initiators [29,30]. The characterization and properties of the prepared nanocomposites were studied using different techniques. The effects of different parameters were studied to estimate the best conditions for the adsorption of methyl blue dye (MB), Figure 1. that was included, the pH of dye solution, the adsorbent dose, and the temperature of the feed solution.

## 2. Materials and Methods

### 2.1. Materials

Pectin (from citrus fruits, galacturonic acid content 93.5%, methoxy content 9.4%, Sigma-Aldrich, St. Louis, MO, USA) and acrylic acid (AAc) of purity 99.9% (Sigma-Aldrich, St. Louis, MO, USA) were used. Other chemicals, such as buffers and dye were purchased from El-Nasr Co. for Chemical Industries, Nasr City, Egypt and used without further purification.

### 2.2. Instrumentations

Microscopic FTIR spectra were investigated by FTIR spectrophotometer, Spectrum One, PerkinElmer, Waltham, MA, USA, over the range of 4000–400 cm^−1^ at a scan rate of 1 spectrum/s at 4 cm^−1^ resolution. XRD analyses were performed by an X-ray diffractometer (XRD, Shimadzu XD-l) with a Cu Ka radiation with (λ = 1.542 Å) at 40 kV operating voltage and 30 mA electric current, at a scanning speed of 8°/min over 2θ of range 4° to 90°.

The topography of the nanocomposite hydrogels was monitored via the AFM, FlexAxiom Nanosurf C3000 at the dynamic mode (non-contact) to confirm chemical modification and to detect the changes accompanying the swelling process. The AFM measurements were conducted at room temperature using an NCLR rectangular-shaped silicon cantilever with a resonant frequency of 9 kHz.

The particle size analysis and zeta potential of CNCs were characterized using dynamic light scattering (DLS, Zetasizer, Malvern Instruments Ltd., Malvern, UK). The prepared CNC sample was dispersed in DI water and ultrasonicated for 15 min under cooling before testing.

The Raman spectrum of CNCs was investigated by Sentra–Bruker, Ettlingen, Germany, in the range of 4000 to 400 cm^−1^ and excitation wavelength 532 nm.

### 2.3. Preparation of Cellulose Nanocrystals (CNCs)

Figure 1 represents the steps of the preparation of CNCs. Pea peels were repeatedly washed and subsequently dried at room temperature ~30–35 °C for three days. Dried pea peels were pulverized and sieved through a 40–60 mm mesh screen. Then, 50 g of powder was put in 4% (*w*/*v*) NaOH for 2 h and filtered. The obtained cellulose was bleached using hydrogen peroxide (30 % *v*/*v*) (300 mL) in a 1 L flask in autoclave 120 °C for an hour. Hydrogen peroxide was added with stirring at 90 °C for 30 min to remove lignin and hemicellulose. The residue was filtered and washed with distilled water to reach neutrality. A total of 30 mL of sulfuric acid (60 % *v*/*v*) was added to the dried cellulose where the ratio of sulfuric acid to cellulose ratio was 1:25 for 30 min. The hydrolysis process was stopped by adding ten folds of ice-cold distilled water to the mixture. The obtained suspension was centrifuged for 30 min at 6500 rpm. It was then washed with distilled water several times and centrifuged until neutralization. Lastly, the neutralized colloidal suspension was sonicated for 10 min to homogenize the generated nano-cellulose.

### 2.4. Preparation of Pectin-PAAc/CNC Nanocomposite

Pectin-PAAc/CNC nanocomposites were prepared by dissolving 5 g of pectin in 80 mL of distilled water at 50 °C for 30 min after cooling. A total of 15 mL of AAc was added with continuous stirring for 2 h at room temperature to obtain a total concentration of 20 wt%. The solution was divided into five equal portions. A definite content of cellulose nanocrystals (0.25, 0.75, 1.25, and 2.0 wt% of the total polymer concentration) was added to each portion. Then the mixtures were sonicated in a bath sonicator for 15 min. They were then transferred into small glass vials and were subjected to the cobalt-60 (60Co) gamma cell at irradiation dose of 20 kGy. This source was installed at the National Center for Radiation Research and Technology (NCRRT), Nasr City, Cairo, Egypt. The nanocomposites were obtained in long cylindrical shapes and were cut into small pieces. All samples were soaked in excess water at 70 °C in a water bath for 24 h to remove the unreacted component, then air-dried at ambient temperature to a constant weight.

### 2.5. Swelling Properties

Pectin-PAAc hydrogels and Pectin-PAAc/CNC nanocomposite hydrogels of known weights were immersed in distilled water at definite interval times until the equilibrium. The swollen samples were re-weighed after the excess surface water was removed immediately with a filter paper. The degree of swelling was determined according to the following equation:(1)$$\mathrm{Equilibrium}\;\mathrm{swelling}\;(\%)=\left(\frac{W_{e}-W_{d}}{W_{d}}\right)\times 100$$
where W_e_ is the weight of the sample after swelling for 24 h, and W_d_ is initial weight of the dry sample.

### 2.6. Adsorption Study

Batch adsorption experiments were carried out at different temperatures: 25, 35, and 50 °C. Exactly 0.1 g of dried composite was put in 20 mL of a known initial concentration of MB solution at pH 9 and was shaken at the agitation speed of 250 rpm for a definite time. The absorption capacity q_e_ of the nanocomposite was calculated using the following Equation:(2)$$q_{e}\left(\mathrm{mg}/g\right)=\frac{\left(C_{0}-C_{t}\right)V}{m}$$
where C_0_ and C_t_ (both in mg/L) are the initial dye concentration and the dye concentration at a definite time, respectively. The MB dye concentration was calculated using a UV/VIS spectrometer, the Jasco model V-530 (Jasco International Co., Ltd., Tokyo, Japan) at λ_max_ of 664 nm with a quartz cell of 1.0 cm optical length.

## 3. Results and Discussion

### 3.1. Characterization of the Prepared CNC

The Raman spectra of CNCs are shown in Figure 2a. The characteristic band of cellulose appears at 1037 cm^−1^ and was assigned to the C–O ring stretching modes and the β-1,4 glycosidic linkage (C–O–C) stretching modes between the glucose rings of the cellulose chains [31]. The average particle size of CNCs was analyzed by DLS as shown in Figure 2b. It was found that the average particle size of CNCs is 23.05 nm, and the zeta potential is −30.7 mV, Figure 2c. The negative value of the zeta potential was attributed to the sulfate anions that are responsible for the repulsion between CNCs and making them stable and well-suspended in an aqueous medium [32]. Figure 2d shows that the XRD diffraction pattern of CNCs has three crystalline peaks at 2θ = 18°, 22°, and 34° corresponding to the CNC-like structure at (110), (200), and (004), respectively, which is in good agreement with the characteristic peaks of cellulose-I structure obtained by Manisha Thakur et al. [33].

AFM was conducted to examine the surface morphology of the purified CNCs. The data were recorded as height image and 3D image, as illustrated in Figure 3. The 3D AFM image (Figure 3a) for the CNCs depicts two distinct regions, bright and dark. The bright region represents the higher crystalline region, while the darker region depicts the lower one [34]. The height of nanocrystals as recorded from the height image is around 3.83 nm. It is evident from Figure 3a that CNCs are well-formed. On the other hand, the crystalline structure of CNCs indicated that the nanocrystals are porous in nature, Figure 3b [35,36]. The 3D image was further analyzed by Gwyddion version 2.60, Czech Metrology Institute, Brno, Czech Republic, Figure 3c, to show the crystalline CNCs in a deeper view.

### 3.2. Fourier Transform Infrared Spectroscopy Analysis (FTIR)

Figure 4 shows the FTIR spectra of CNCs, pectin, Pectin-PAAc, and Pectin-PAAc/CNC nanocomposite. For CNCs, the band of –OH appears around 3351 cm^−1^. The weak band at 1640 cm^−1^ was attributed to the O–H vibration of absorbed water [37]. The band at 1064 cm^−1^ was assigned to the stretching vibration of C–O–C of the pyranose ring. For pectin, a broad band at 3370 cm^−1^ was attributed to O–H. The C–H asymmetrical stretching of the –CH_2_ groups appears at 2944 cm^−1^. The band at 1710 cm^−1^ was assigned to C=O. The band at 1010 cm^−1^ is due to C–O of glycosidic bonds [38]. For the Pectin-PAAc hydrogel, a very strong, sharp band appears at 1691 cm^−1^, which was assigned to the asymmetrical stretching vibration of the carbonyl group of -COOH of pectin and AAc. This band is accomplished by the appearance of the symmetrical stretching vibration band at 1455 cm^−1^ [39]. The band at 1158 cm^−1^ is due to the stretching vibration of –C–O of the carboxylic moiety. The band at 794 cm^−1^ was attributed to the rocking vibration of C–H. For the Pectin-PAAc/CNC nanocomposite, no new bands appear, but the intensity of the carbonyl band at 1691 cm^−1^ is diminished and negatively shifted. This may be attributed to the enhancement and reorientation of the hydrogen bonding in the nanocomposite matrix by including CNCs in the structure [40]. Furthermore, no evident absorption peaks appeared after the addition of CNCs in the Pectin-PAAc/CNC nanocomposite hydrogels, suggesting that the incorporation of CNCs did not change the chemical structure of the Pectin-PAAc hydrogel. Hence, it can be said that the reinforcement of Pectin-PAAc hydrogels by CNCs was mainly connected to the physical crosslinking, electrostatic interaction, and hydrogen bonding between the CNC filler and the Pectin-PAAc matrix [41].

### 3.3. X-ray Diffraction Analysis (XRD)

Figure 5 shows the XRD diffractograms of Pectin-PAAc and the Pectin-PAAc/CNC nanocomposite. The diffractograms of both Pectin-PAAc and the Pectin-PAAc/CNC nanocomposite showed broad peaks centered at 2θ = 23.7°, reflecting the amorphous structure of Pectin-PAAc and the Pectin-PAAc/CNC nanocomposite. The intensity of the peak of the Pectin-PAAc/CNC nanocomposite is lower than Pectin-PAAc due to the presence of CNCs. On the other hand, it must be observed that the crystalline peaks of CNCs do not appear in the Pectin-PAAc/CNC nanocomposite due to the low concentration of CNCs in the nanocomposite matrix.

### 3.4. Swelling Behavior

The swelling behavior of the Pectin-PAAc hydrogel and the Pectin-PAAc/CNC nanocomposite, containing 0.125 wt% CNCs, in different pH media (3, 5, 7, 9, and 11), is given in Figure 6. It can be recognized that both systems have a high swelling capability. However, the Pectin-PAAc/CNC nanocomposite has higher swelling (%) than the Pectin-PAAc hydrogel. The swelling percentage gradually increases as the pH value increases, to get the maximum value at a pH of 11, where the swelling percentages of Pectin-PAAc and Pectin-PAAc/CNC are 7343 and 10,281, respectively. It must be noted that Pectin-PAAc and Pectin-PAAc/CNC display the lowest swelling percentages of 1027 and 1236 at pH 3, respectively. This is mainly due to the acceleration of hydrogen bonding formation by the protonation of the anionic AAc in the acidic medium, which restricts the swelling capability of nanocomposite hydrogel. As the pH increases, the deprotonation of the carboxylic groups of AAc also increases. Therefore, improving the repulsive force between the inter-macromolecular chains in the matrix leads to the expansion of the interstitial space for holding water so that the swelling ratio gradually increases [42].

The inclusion of CNCs in the nanocomposite matrix seemed to have a significant effect on the swelling performance of the Pectin-PAAc/CNC nanocomposite hydrogel as seen in Figure 7. The swelling of the parent Pectin-PAAc hydrogel is 4507%, while the inclusion of 0.025, 0.075, and 0.125 wt% of CNCs in the matrix has improved the swelling to 5490, 6699, and 9634%, respectively. This means that the presence of 0.125 wt% of CNCs in the Pectin-PAAc/CNC nanocomposite hydrogel duplicates the swelling percentage due to the hydrophilic nature of CNCs. The presence of CNCs enhances the matrix swelling [43]. However, the increase of CNC content to 0.2 wt% dropped the swelling percentage to 2294%. This observation was mostly attributed to the improvement in the cross-linking of the Pectin-PAAc/CNC nanocomposite. The high content of CNCs narrowed the voids in the nanocomposite network and rendered the structure denser, thus reducing the swelling performance [44]. Overall, the results illustrated that the inclusion of CNCs exerted a significantly positive effect on the swelling performance of the resulting nanocomposite hydrogels.

### 3.5. Dye Adsorption

The influence of MB dye concentration was studied at three temperatures: 298, 308, and 318 K, and the results are presented in Figure 8. The adsorption capacity increases with increasing the adsorbate concentration (MB) up to 70 mg/L where the highest adsorption capacity for both systems at all investigated temperatures was achieved. Increasing the MB concentration enhanced the adsorption capacity by improving the contact between the MB molecules and active sites. Improving the initial dye concentration enhances the driving force to overcome the resistance of the mass transfer of dye between the solute and the adsorbent surface. At a low concentration of MB, a high rate of moving dye molecules to the vacant available position on the adsorbent is observed. This availability reduces as the MB concentration increases, until equilibrium, where the active sites become occupied [45]. On the other hand, the temperature significantly affects the adsorption capacity. The increase in temperature enhances the adsorption capacity. The results indicated that the adsorption process of MB onto the nanocomposite hydrogel possesses an endothermic nature.

### 3.6. Adsorption Kinetics

The adsorption kinetics provide information for designing and modeling the adsorption process. The mechanism that controls the adsorption process (rate-determining step) of MB onto Pectin-PAAc and Pectin-PAAc/CNC was investigated by fitting the experimental kinetics data by adsorption kinetics models, such as pseudo-first-order (Equation (3)), pseudo-second-order (Equation (4)), and Avrami (Equation (5)) kinetic models [46]. Figure 9 shows the kinetic adsorption curves for the adsorption of MB by Pectin-PAAc and Pectin-PAAc/CNC using the investigated adsorption kinetic models. The analyzed adsorption kinetic parameters are listed in Table 1.
(3)$$q_{t}=q_{e}\left(1-e^{-k_{1}t}\right)$$
(4)$$q_{t}=\frac{k_{2}q_{e}^{2}t}{1+k_{2}q_{e}t}$$
(5)$$q_{t}=q_{e}\left(1-\exp{\left[-k_{\mathrm{AV}}t\right]}^{n_{\mathrm{AV}}}\right)$$
where k_1_ is the PFO adsorption rate constant (min^−1^); k_2_ is the PSO adsorption rate constant (g mg^−1 ^min^−1^); q_e_ and q_t_ are the amounts of MB adsorbed per g of biosorbent at equilibrium and at a time, t, respectively, in (mg g^−1^); t is time (min); k_AV_ is Avrami’s kinetic constant (min^−1^); and n_AV_ is the fractional order of the reaction, which is related to the mechanism adsorption.

To compare the validity of model Equations more definitely a normalized standard deviation (%) is calculated as follows:(6)$$∆q\left(\%\right)=100\times \sqrt{\frac{{\left[\left(q_{e}^{\exp}-q_{e}^{\mathrm{cal}}\right)/q_{e}^{\exp}\right]}^{2}}{N-1}}$$
where the superscripts “exp” and “cal” show the experimental and calculated values, and N is the number of measurements.

It can be observed that the correlation coefficients of the three kinetic models applied to Pectin-PAAc/CNC adsorption are higher than 0.97. It was estimated that the kinetic models fit the adsorption of MB well. So, their applicability can be based on the value of ∆qe; the Avr kinetic model has the lowest one. Thus, the Avr model is the most suitable model for describing the adsorption kinetics. Furthermore, the q_e_ (cal) is the nearest to the q_e_ (exp). The Avr exponential is a fractional number related to the possible changes in the adsorption mechanism during the adsorption process. Therefore, the adsorption mechanism can follow multiple kinetic orders that may change during the connection between the adsorbate and the adsorbent. For the Pectin-PAAc hydrogel, the R^2^ value of the PFO model is the highest. The ∆q_e_ (%) value for the PFO model is lower than the obtained value for the PSO model. Based on this result, the PFO model offered the best fit for the experimental data. It possessed the highest correlation coefficient, and the q_exp_ value agreed with the calculated one. This means the adsorption process is controlled by diffusion through the interface. Thus, the mechanism that controls the adsorption process of MB by Pectin-PAAc hydrogel changed by the inclusion of CNCs in the nanocomposite matrix of Pectin-PAAc/CNC.

### 3.7. Intraparticle Diffusion

The adsorption rate is mainly controlled by the first two steps because the last one is very fast. Intraparticle diffusion is the rate-limiting factor in the adsorption process if the plot of the adsorbed amount against the square root of the contact time yields a straight line. The intraparticle diffusion Equation for the adsorption system is given by:(7)$$q_{t}=k_{i}t^{0.5}+C$$
where q_t_ is the adsorption capacity at time t, K_i_ (mg g^−1^ min^1/2^) is the rate constant of intraparticle diffusion, and C is the intercept (mg/g).

The parameters of the intraparticle diffusion model are presented in Table 2. The intraparticle diffusion plot (Figure 10) shows multilinearity, indicating that the adsorption process is composed of several stages. Thus, intraparticle diffusion is not the only rate-controlling step in the adsorption process.

### 3.8. Adsorption Isotherm Models

The adsorption isotherm model is very useful to explain the interaction between the adsorbate and the adsorbent of any system. It indicates the affinity of the adsorbent towards an adsorbate. The isotherm study is conducted by studying the relation between the adsorbate concentration at equilibrium in the liquid phase and the solid phase at a definite temperature. Several models can analyze the experimental adsorption equilibrium data to obtain the known adsorption as a physical or chemical process [47] and the most accepted surface adsorption model [48]. On the other hand, the parameters obtained from the different models give information about the adsorption mechanism and the surface properties.

In this study, two-parameter adsorption isotherm models (Langmuir Equation (8) [49], Freundlich Equation (9) [50], Jovanovic Equation (10) [51], and Temkin Equation (11) [52]) and three-parameter adsorption isotherm models (Redlich–Peterson [53] Equation (12) and Brouers–Sotolongo Equation (13) [54]) were applied to the investigated systems as shown in Figure 11. The parameters of these isotherms are shown in Table 3. All the models were fit employing the non-linear fitting method by the OriginPro 2016 software, OriginLab Corporation, Northampton, MA, USA.

The Langmuir isotherm model is based on monolayer adsorption. The adsorbent surface is homogeneous and controlled only by one type of binding site. It assumes a negligible interaction between the adsorbate molecules, represented as:(8)$$q_{e}=\frac{q_{m}K_{L}C_{e}}{1+K_{L}C_{e}}$$
where C_e_ (mg/L) is the equilibrium adsorbate concentration, q_e_ (mg/g) is the equilibrium adsorbed amount of adsorbates, K_L_ is the Langmuir constant (L/mg), and q_m_ is the adsorbent’s maximum adsorption capacity (mg/g).

Figure 11a shows a great fit with the experimental data where R^2^ values are 0.9971 and 0.9999 for Pectin-PAAc and Pectin-PAAc/CNC, respectively, close to unity. The obtained q_m_ value is 486.67 mg/g for Pectin-PAAc and 576.62 mg/g for Pectin-PAAc/CNC.

The Freundlich isotherm model describes the adsorption of heterogeneous surfaces as well as the multilayer adsorption, represented as:(9)$$q_{e}=K_{F}C_{e}^{1/n}$$
where K_F_ is the Freundlich isotherm constant and n is the adsorption intensity. The adsorption is considered a favorable process when the n value is greater than unity.

The R^2^ values are lower than the Langmuir isotherm model values (Table 3). The values of n are higher than 1 for both systems, which indicates the adsorption process is a favorable process.

The Temkin isotherm model considers the adsorbate’s interactions, like the Freundlich isotherm, and assumes that the adsorption heat of all molecules reduces linearly as the layer is wrapped. It is characterized by a uniform distribution of binding energies up to maximum energy, represented as:(10)$$q_{e}=\frac{\mathrm{RT}}{b_{T}}\ln\;\left(K_{T}C_{e}\right)$$
where R is the universal gas constant (8.314 J/mol K), T (K) is absolute temperature, K_T_ (L/g) is the Temkin constant, and b_T_ (J/mol) is a constant related to the heat of adsorption.

It can be observed from Table 3 that this isotherm model has the highest ∆q_e_ values for both systems, which means it is unsuitable for these adsorption systems. On the other hand, the values of b_T_ are positive, indicating the exothermic adsorption process.

The Jovanovic isotherm model describes the chemical adsorption of moderate concentration of the adsorbate. It is in agreement with the Langmuir isotherm model with the permeation of mechanical interaction, and is represented as:(11)$$q_{e}=q_{\mathrm{mJ}}\times \left(1-e^{K_{J}C_{e}}\right)$$
where K_J_ (L/mg) is the Jovanovich constant and q_mJ_ (mg/g) is the Jovanovich maximum adsorption capacity.

The lower values of R^2^ compared with the Langmuir isotherm model may be due to the weak mechanical interaction.

The Redlich–Peterson isotherm is applied to homogeneous and heterogeneous systems. It fits between the Langmuir and the Freundlich models, represented as:(12)$$q_{e}=\frac{K_{\mathrm{RP}}C_{e}}{1+a_{R}C_{e}^{b_{\mathrm{RP}}}}$$
where $K_{\mathrm{RP}},\;a_{R}\;\mathrm{and}\;b_{\mathrm{RP}}$ are Redlich–Peterson’s constants.

The value of $b_{\mathrm{RP}}$ is considered preferable for the system. If $b_{\mathrm{RP}}$ is equal or closed to 1, the isotherm becomes identical with the Langmuir isotherm. If $b_{\mathrm{RP}}$ is equal or closed to 0, the isotherm becomes identical with the Freundlich isotherm. The values of $b_{\mathrm{RP}}$ are 0.9697 and 0.9713 for Pectin-PAAc and Pectin-PAAc/CNC, respectively. It means the value of $b_{\mathrm{RP}}$ is close to the unit indicated that the Langmuir isotherm is more suitable for them.

The Brouers–Sotolongo isotherm model is represented as the following:(13)$$q_{e}=q_{\mathrm{mBS}}(1-\exp(-K_{\mathrm{BS}}{\left(C_{e}\right)}^{\alpha_{\mathrm{BS}}}))$$
$K_{\mathrm{BS}}$ and $\alpha_{\mathrm{BS}}$ are the Brouers–Sotolongo constants and $q_{\mathrm{mBS}}$ (mg/g) is the Brouers–Sotolongo maximum adsorption capacity.

The $q_{\mathrm{mBS}}$ values are 468.73 and 572.65 mg/g for Pectin-PAAc and Pectin-PAAc/CNC, respectively, which are close to the Langmuir isotherm values.

It must be noted that the most appropriate isotherm model that describes the experimental data of both Pectin-PAAc and Pectin-PAAc/CNC is chosen according to correlation coefficient (R^2^) values and the normalized standard deviation (∆qe). After analyzing all the isotherms and calculating the parameters, it can be noted that the descending arrangement of isotherm suitability according to R^2^ for Pectin-PAAc hydrogel is Brouers–Sotolongo > Langmuir > Redlich–Peterson > Temkin > Freundlich > Jovanovic. For Pectin-PAAc/CNC nanocomposite the order is Langmuir > Freundlich > Brouers–Sotolongo > Redlich–Peterson > Temkin > Jovanovic.

On the other hand, the ascending arrangement of isotherm suitability according to ∆q_e_ for Pectin-PAAc hydrogel is Brouers–Sotolongo > Langmuir > Jovanovic > Redlich–Peterson > Freundlich > Temkin. For Pectin-PAAc/CNC nanocomposite the order is Langmuir > Jovanovic > Brouers–Sotolongo > Redlich–Peterson > Freundlich >Temkin.

According to the R^2^, ∆qe, and qm, it can be concluded that the adsorption isotherm of MB by Pectin-PAAc follows the Brouers–Sotolongo model, which is close to the Langmuir model. The adsorption isotherm of MB by Pectin-PAAc/CNC follows the Langmuir isotherm model. This means the adsorption of MB by Pectin-PAAc and Pectin-PAAc/CNC is mainly chemical adsorption.

### 3.9. Thermodynamic Parameters of Adsorption

The adsorption thermodynamics were carried out to obtain information about the internal energy and structure change after adsorption. It is valuable to understand the spontaneity and the effect of heat on adsorption. The thermodynamics of adsorption of MB by Pectin-PAAc and Pectin-PAAc/CNC nanocomposite hydrogels were performed at 298 K, 308 K, and 318 K. The thermodynamic parameters, such as enthalpy change (∆H), entropy change (∆S), and Gibbs free energy change (∆G) can be calculated by the following Equations:(14)$$\ln\frac{q_{e}}{C_{e}}=\frac{∆S}{R}-\frac{∆H}{\mathrm{RT}}$$
(15)∆G=∆H−T∆S
where T is the absolute temperature (K) and R is the universal gas constant (8.314 J mol^−1^K^−1^). The values of ∆H and ∆S can be obtained from the slope and intercept of the linear graph, about $\ln\frac{q_{e}}{C_{e}}$ versus 1/T, as shown in Figure 12, and the calculated parameters are listed in Table 4.

From Table 4 it can be observed that all values of ∆G are negative, while ∆H and ∆S are positive. The Gibbs free energy indicates the spontaneity of the adsorption process, as well as the higher negative value, which suggest more energetically favorable adsorption. The negative values of ∆G confirm the thermodynamic feasibility of the process and spontaneous nature of adsorption [55]. The positive values of ∆H indicate the endothermic nature of the adsorption, so the energy is required for the reaction to occur [56]. The positive values of ∆S reveal the increase in randomness for solid-solution interface and a good affinity of the adsorbent material to absorb MB dye [57]. On the other hand, the change in the concentration of MB affected the thermodynamic parameters. The value of ∆S increases by 0.4 and 0.9 (kJ mol^−1^ K^−1^) when the the MB concentration is increased from 50 to 70 mg/L for Pectin-PAAc and Pectin-PAAc/CNC, respectively, due to the increasing affinity of MB towards the adsorbent. The randomness increases at the adsorbent/adsorbate interface through the adsorption process. It can be deduced that MB molecules on the adsorbent surface substitute more water molecules. However, increasing the concentration from 70 to 100 mg/L decreases ∆S for both systems by increasing the completion between the MB molecules over the interaction on the adsorbent surface. Increasing the magnitude of ∆G to a higher temperature at a constant MB concentration reflected increasing the feasibility of the process by raising the temperature. Nevertheless, this feasibility and spontaneity of the adsorption process diminished by increasing the MB concentration.

### 3.10. Atomic Force Microscopy (AFM)

The surface topography of Pectin-PAAc and Pectin-PAAc/CNC nanocomposites was examined by AFM. The topography was investigated for dry, de-swelled Pectin-PAAc and Pectin-PAAc/CNC nanocomposites and after the MB adsorption process, as shown in Figure 13. A rough surface was observed for dried Pectin-PAAc hydrogel, with a height of 153 nm while a shrinkage of the thickness of the de-swelled sample of 115 nm after immersion for 24 h in distilled water and lyophilization. The surface completely changed after the adsorption of MB dye: the height increased to 294 nm due to the presence of MB dye adsorbate, reflecting the effective adsorption of the Pectin-PAAc hydrogel. On the other hand, the topography image of dry Pectin-PAAc/CNC nanocomposite appears as a rough surface, like the Pectin-PAAc hydrogel, with little surface change and a height of 154 nm due to the incorporation of CNCs in Pectin-PAAc. The high compatibility and distribution of CNCs within the polymeric matrix is clear. Increased roughness was observed for the de-swelled nanocomposite sample, with an increased height of 181 nm. This may be due to the high swelling of the nanocomposite where the surface keeps the pore cavities after de-swelling. The surface changes after the adsorption of MB highly enhanced the height to about three times that of the dry sample, 474 nm. It can be concluded that the presence of CNCs influences the surface topography of Pectin-PAAc/CNC nanocomposites. Furthermore, after MB adsorption, the surface height of Pectin-PAAc/CNC nanocomposite is nearly twice that of Pectin-PAAc. This was explained by the clear influence of the incorporation of CNCs in the composite matrix of Pectin-PAAc/CNC nanocomposite.

## 4. Conclusions

CNCs were successfully produced from pea peels by acid hydrolysis, with an average particle size of 23.05 nm, and a zeta potential of –30.7 mV. CNCs were used in the formation of a Pectin-PAAc/CNC nanocomposite by γ-irradiation. FTIR confirmed that the incorporation of CNCs did not change the chemical structure of the Pectin-PAAc hydrogel. The presence of CNCs in the polymeric matrix enhances the swelling and adsorption properties of Pectin-PAAc/CNC. The swelling percentage increases as the pH value increases, to reach a maximum pH value of 11. The adsorption capacity increases with an increasing MB concentration, up to 70 mg/L. It was found that the kinetic models fit the adsorption of MB well where the correlation coefficients of all kinetic models are higher than 0.9700. The Avr kinetic model has the lowest ∆q_e_ value, making it the most suitable for describing the adsorption kinetics. The intraparticle diffusion was found not to be the only rate-controlling step in the adsorption process. The descending arrangement of isotherm suitability according to the R^2^ for Pectin-PAAc hydrogel is Brouers–Sotolongo > Langmuir > Redlich–Peterson > Temkin > Freundlich > Jovanovic. For Pectin-PAAc/CNC nanocomposite the order is Langmuir > Freundlich > Brouers–Sotolongo > Redlich–Peterson > Temkin > Jovanovic. The adsorption isotherm of MB by Pectin-PAAc follows the Brouers–Sotolongo model while that by Pectin-PAAc/CNC follows the Langmuir isotherm model. The negative values of ∆G confirmed the spontaneous nature of adsorption, and the positive values of ∆H indicated the endothermic nature of the adsorption. The AFM investigations confirmed that the presence of CNCs influences the surface topography of Pectin-PAAc/CNC nanocomposites. After MB adsorption, the surface height of Pectin-PAAc/CNC nanocomposite is nearly twice that of Pectin-PAAc.

## Data Availability

The data that support the findings of this study are available from the corresponding author upon reasonable request.

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
