# Peer review of "Engineered Superabsorbent Nanocomposite Reinforced with Cellulose Nanocrystals for Remediation of Basic Dyes: Isotherm, Kinetic, and Thermodynamic Studies"

_polymers, 2022, doi:10.3390/polym14030567_

Round 1

Reviewer 1 Report

The presented manuscript requires additional revision. So the objective of the work needs to be more clearly defined and emphasized. The work does not present diffractograms, how can the authors determine that they have received the crystalline phase without them? The manuscript contains a large number of typos and errors that need to be corrected.

Line 18. "SFO" should be checked, maybe authors have input "PFO"
Line 20. "0.9700" - remove extra zeros.
Line 21. What is "? Qe"? The reader will not understand from the abstract.
DLS - no abbreviation decryption.
Figure 2.b. Do CNC particles swell in the measurement environment?
Line 202 onwards. Round off values ​​"4507.34%".
Figure 5. "CNC cnontent (wt%)" - needs to be fixed.
Figure 6. "Initiail" - needs to be fixed.
"The adsorption capacity increases with increasing MB concentration" - it is necessary to explain how this happens !? 

Author Response

Manuscript ID: polymers-1540782

Title: Engineered superabsorbent nanocomposite reinforced with cellulose nanocrystal for remediation of Basic dyes: Isotherm, kinetic, and thermodynamic studies

 Response to the Referee’s report

 We would like to thank the reviewers for the carefully reading of our manuscript and for giving such constructive comments which substantially helped improving the quality of the paper. Guided by these comments and suggestions, we have made careful modifications to meet all comments and suggestions. The points raised by the reviewer are repeated (in bold letters) followed by the authors’ reply.  

Response to Reviewers

“The work does not present diffractograms, how can the authors determine that they have received the crystalline phase without them?”

To address this point, we have done XRD diffractograms for CNC presented in Fig2.  the XRD diffractograms of Pectin-PAAc and Pectin-PAAc/CNC nanocomposite presented in Fig3b.  

“The manuscript contains a large number of typos and errors that need to be corrected”

Line 18. "SFO" should be checked, maybe authors have input "PFO", Line 20. "0.9700" - remove extra zeros. Line 21. What is "? Qe"? The reader will not understand from the abstract. DLS - no abbreviation decryption. Figure 2.b. Do CNC particles swell in the measurement environment? Line 202 onwards. Round off values "4507.34%". Figure 5. "CNC cnontent (wt%)" - needs to be fixed. Figure 6. "Initiail" - needs to be fixed.

Thank you for pointing out this comment, we have checked these typos in a revised manuscript.

"The adsorption capacity increases with increasing MB concentration" - it is necessary to explain how this happens !? 

√ done.  Lines 259-261 page 10

Once again, we do appreciate very much the constructive your comments to improve our manuscript and we hope you finds the paper more convincing.

Reviewer 2 Report

Dear authors,

This manuscript deals with a superabsorbent nanocomposite reinforced with cellulose nanocrystal for remediation of Basic dyes: Isotherm, kinetic, and thermodynamic studies.

Several items must be attended to before it can be considered for publication.

Line 34... 3X105... reduce the size of X.

The citation must be placed uniformly along with the manuscript... word..space...cite.

Line 56-57. make a complete definition abut the chemical composition of pectin, including methylation and sugar units. What is the advantage of using it?

Line 67... complete the sentence idea of reference 26, the purpose of the biocomposite ? Is crystallinity an interesting property for this paper purpose.?..It is necessary to present other references where CNC had been previously used for composites preparation used in dyes absorption, for instance. There are several of them.

The same comment for previous pectin/PAA composite preparation.

Methodology section

Line 86... FTIR number of scans must be placed... the same in Line 96 with Raman spectrum

Line 118...replace the symbol &...with word

Line 120 (Gamma Co supplier?)

Line 124...air dried ...at which temperature?

Line 138...remove ; after pH

Results section

Line 151...cm-1...upper case -1

Better discussion and comparison with previous papers of each parameter must be added for each section of the manuscript.

In figure 1...the pectin FTIR is necessary to be included. discussion and comparison are necessary.

What is the reason behind the higher deltaS values at 70 mg/L than 50 and 100 mg/L, respectively?

It is suggested to include other techniques for composite characterization like TGA , SEM and or XRD for biocomposite characterization

Also, SEM, TEM or AFM picture of CNC is mandatory to demonstrate you are really working with nanocrystals and not with nanofibers. Also, SEM analysis of biocomposite can be add

Author Response

Manuscript ID: polymers-1540782

Title: Engineered superabsorbent nanocomposite reinforced with cellulose nanocrystal for remediation of Basic dyes: Isotherm, kinetic, and thermodynamic studies

 Response to the Referee’s report

 We would like to thank the reviewers for the carefully reading of our manuscript and for giving such constructive comments which substantially helped improving the quality of the paper. Guided by these comments and suggestions, we have made careful modifications to meet all comments and suggestions. The points raised by the reviewer are repeated (in bold letters) followed by the authors’ reply.  

Response to Reviewers

Dear authors,

This manuscript deals with a superabsorbent nanocomposite reinforced with cellulose nanocrystal for remediation of Basic dyes: Isotherm, kinetic, and thermodynamic studies. Several items must be attended to before it can be considered for publication.

“Line 34... 3X105... reduce the size of X” , “The citation must be placed uniformly along with the manuscript... word..space...cite”

Thank you for pointing out this comment, we have checked these typos in a revised manuscript.

Line 56-57. make a complete definition about the chemical composition of pectin, including methylation and sugar units.

To address this point, we add a new paragraph Lines 58-60 page 2.

What is the advantage of using it?

Pectin is a non-toxic, biodegradable, and biocompatible polymeric material. Line 55

Line 67... complete the sentence idea of reference 26, the purpose of the biocomposite? Is crystallinity an interesting property for this paper purpose.?..It is necessary to present other references where CNC had been previously used for composites preparation used in dyes absorption, for instance. There are several of them.

√ done. 

In figure 1...the pectin FTIR is necessary to be included. discussion and comparison are necessary.

To address this comment, in the revised manuscript we modified figure 3a, including Pectin FTIR spectrum and rewrite FTIR sec.

What is the reason behind the higher deltaS values at 70 mg/L than 50 and 100 mg/L, respectively?

The importance is this change is negative or positive value. The same attitude is observed in the following references and the authors concerned with the sign of delta S is positive or negative.

Sulaiman, N. S., Mohamad Amini, M. H., Danish, M., Sulaiman, O., & Hashim, R. (2021). Kinetics, Thermodynamics, and Isotherms of Methylene Blue Adsorption Study onto Cassava Stem Activated Carbon. Water, 13(20), 2936.‏

Baskaran, K., Venkatraman, B. R., Hema, M., & Arivoli, S. (2011). Adsorption kinetics and thermodynamics of malachite green dye onto Calatropis gigantis bark carbon. I Control Pollution, 27(1)

“Methodology section, Line 86... FTIR number of scans must be placed... the same in Line 96 with Raman spectrum, Line 118...replace the symbol &...with word, Line 120 (Gamma Co supplier?), Line 124...air dried ...at which temperature?,  Line 138...remove ; after pH, Results section ,Line 151...cm-1...upper case -1”

Thank you for pointing out this comment, we have checked in a revised manuscript.

It is suggested to include other techniques for composite characterization like TGA , SEM and or XRD for biocomposite characterization

Your suggestion is well received, actually we believe shedding the light on this comment improves the quality of the manuscript, in addressing this issue we have included the XRD diffractograms of Pectin-PAAc and Pectin-PAAc/CNC nanocomposite (Figure 3b).  We have discussed this point in the revised manuscript on (page 8).

Also, SEM, TEM or AFM picture of CNC is mandatory to demonstrate you are really working with nanocrystals and not with nanofibers. Also, SEM analysis of biocomposite can be add

We totally agree with the reviewer. Never the less, we can’t able in the present time we hope that the reviewer would beer with us and accept our plea.

Once again, we do appreciate very much the constructive your comments to improve our manuscript and we hope you finds the paper more convincing.

Round 2

Reviewer 1 Report

The authors made a number of corrections and took into account the previous comments, but not in full. The manuscript is overloaded with values ​​that need to be rounded and contains a large number of typos. Below are examples of errors that need to be corrected before posting:

Line 38. fix "[5, 4]"
Line 40. ";" replaced by ":"
Line 94. In my opinion, the not entirely successful statement "The infrared spectra were investigated by FTIR spectrophotometer"
Lines 106, 107. Delete - "serial number: MAL1071664" and "ver. 6.32"
Line 117. Fix - "60-40".
Line 129. Replace "Figure (1): the" with "Figure 1. The"
Line 134. ", nanorystal" replace with "nanocrystals"
Line 135. "0.0" - delete.
Lines 136, 137. "The mixtures were sonicated in a bath sonicator for 15 min to obtain homogenous solutions." - not a correct statement.
Line 175. "2θ = 18 °" - you need to check the angular position for 110.
Figure 2. Swap figures c and d.
Lines 194-196. Can the authors confirm this statement with a link?
Line 214. "shorter" - replace.
Line 224 onwards. "7342.57 and 10281.39" - round off. 

Author Response

Manuscript ID: polymers-1540782

Title: Engineered superabsorbent nanocomposite reinforced with cellulose nanocrystal for remediation of Basic dyes: Isotherm, kinetic, and thermodynamic studies

 Response to the Referee’s report

 We would like to thank the reviewers for the carefully reading of our manuscript and for giving such constructive comments which substantially helped improving the quality of the paper. Guided by these comments and suggestions, we have made careful modifications to meet all comments and suggestions. The points raised by the reviewer are repeated (in bold letters) followed by the authors’ reply.  

Response to Reviewer

The authors made a number of corrections and took into account the previous comments, but not in full. The manuscript is overloaded with values ​​that need to be rounded and contains a large number of typos. Below are examples of errors that need to be corrected before posting:

Line 38. fix "[5, 4]"
Line 40. ";" replaced by ":"
Line 94. In my opinion, the not entirely successful statement "The infrared spectra were investigated by FTIR spectrophotometer"
Lines 106, 107. Delete - "serial number: MAL1071664" and "ver. 6.32"
Line 117. Fix - "60-40".
Line 129. Replace "Figure (1): the" with "Figure 1. The"
Line 134. ", nanorystal" replace with "nanocrystals"
Line 135. "0.0" - delete.
Lines 136, 137. "The mixtures were sonicated in a bath sonicator for 15 min to obtain homogenous solutions." - not a correct statement.
Figure 2. Swap figures c and d.
Lines 194-196. Can the authors confirm this statement with a link?
Line 214. "shorter" - replace.
Line 224 onwards. "7342.57 and 10281.39" - round off. 

Thank you for pointing out these comments, we have checked these typos in a revised manuscript.

Lines 194-196. Can the authors confirm this statement with a link?

√ done

Once again, we do appreciate very much the constructive your comments to improve our manuscript and we hope you finds the paper more convincing.

Reviewer 2 Report

Dear authors,

The SEM, TEM or AFM picture is mandatory in this case to really demonstrate you really obtained cellulose nanocrystals .

Author Response

Manuscript ID: polymers-1540782

Title: Engineered superabsorbent nanocomposite reinforced with cellulose nanocrystal for remediation of Basic dyes: Isotherm, kinetic, and thermodynamic studies

 Response to the Referee’s report

 We would like to thank the reviewers for the carefully reading of our manuscript and for giving such constructive comments which substantially helped improving the quality of the paper. Guided by these comments and suggestions, we have made careful modifications to meet all comments and suggestions. The points raised by the reviewer are repeated (in bold letters) followed by the authors’ reply.  

Response to Reviewers

Dear authors,

The SEM, TEM or AFM picture is mandatory in this case to really demonstrate you really obtained cellulose nanocrystals.

To address this point, we have added AFM images of CNCs (Figure 3).  We have discussed this point in the revised manuscript on (lines 178 to 184).

Once again, we do appreciate very much the constructive your comments to improve our manuscript and we hope you finds the paper more convincing.

Round 3

Reviewer 1 Report

The authors have made a number of edits to the article, below are additional questions and recommendations.
As for the article, for me, first of all, it was not enough to compare the described system with analogues (in particular, with the systems (membranes) used). As a recommendation, I suggest the authors add relevant information to the theoretical part. 

Line 64. I suggest removing "rigid" and replacing "nanoparticle" with "particle".
Line 70. Need to fix "Many studied used"
Line 85. Repetition - "dye (MB) dye"
Line 107, 108. It is important to indicate whether the system was cooled during processing.
Line 116. " dried under the sunlight" - Change to temperature (range).
Line 137. "solutions" could be "suspension" after all?
Lines 174-176. How true is it to say that there is one peak in the region of 18°? Maybe it's an overlap of two peaks? 

Author Response

 Response to the Referee’s report

 We would like to thank the reviewers for the carefully reading of our manuscript and for giving such constructive comments which substantially helped improving the quality of the paper.

Response to Reviewer

Line 64. I suggest removing "rigid" and replacing "nanoparticle" with "particle".
Line 70. Need to fix "Many studied used"
Line 85. Repetition - "dye (MB) dye"
Line 107, 108. It is important to indicate whether the system was cooled during processing.
Line 116. " dried under the sunlight" - Change to temperature (range)

Thank you for pointing out these comments, we have checked these typos in a revised manuscript.

Once again, we do appreciate very much the constructive your comments to improve our manuscript and we hope you finds the paper more convincing.

Reviewer 2 Report

No comments

Author Response

 We would like to thank the reviewer